# The Role of Microbiome in Brain Development and Neurodegenerative Diseases

**DOI:** 10.3390/molecules27113402

**Published:** 2022-05-25

**Authors:** Varsha Nandwana, Nitesh K. Nandwana, Yogarupa Das, Mariko Saito, Tanisha Panda, Sasmita Das, Frankis Almaguel, Narayan S. Hosmane, Bhaskar C. Das

**Affiliations:** 1Arnold and Marie Schwartz College of Pharmacy and Health Sciences, Long Island University, Brooklyn, NY 11201, USA; varshanan3@gmail.com (V.N.); nitesh.nandwana@liu.edu (N.K.N.); pandatanisha69@gmail.com (T.P.); sasmita.das@liu.edu (S.D.); 2Department of Medicine, Icahn School of Medicine at Mount Sinai, New York, NY 10029, USA; 3Nathan S. Kline Institute for Psychiatric Research, Orangeburg, NY 10962, USA; rojadas08@gmail.com (Y.D.); mariko.saito@nki.rfmh.org (M.S.); 4School of Medicine, Loma Linda University Health, Loma Linda, CA 92350, USA; falmaguel@llu.edu; 5Department of Chemistry and Biochemistry, Northern Illinois University, DeKalb, IL 60115, USA; hosmane@niu.edu

**Keywords:** microbiome, gut-brain axis, boron-based diet, Alzheimer’s disease, Parkinson’s disease, boron neuroprotective agent

## Abstract

Hundreds of billions of commensal microorganisms live in and on our bodies, most of which colonize the gut shortly after birth and stay there for the rest of our lives. In animal models, bidirectional communications between the central nervous system and gut microbiota (Gut–Brain Axis) have been extensively studied, and it is clear that changes in microbiota composition play a vital role in the pathogenesis of various neurodevelopmental and neurodegenerative disorders, such as Autism Spectrum Disorder, Alzheimer’s disease, Parkinson’s disease, Multiple Sclerosis, Amyotrophic Lateral Sclerosis, anxiety, stress, and so on. The makeup of the microbiome is impacted by a variety of factors, such as genetics, health status, method of delivery, environment, nutrition, and exercise, and the present understanding of the role of gut microbiota and its metabolites in the preservation of brain functioning and the development of the aforementioned neurological illnesses is summarized in this review article. Furthermore, we discuss current breakthroughs in the use of probiotics, prebiotics, and synbiotics to address neurological illnesses. Moreover, we also discussed the role of boron-based diet in memory, boron and microbiome relation, boron as anti-inflammatory agents, and boron in neurodegenerative diseases. In addition, in the coming years, boron reagents will play a significant role to improve dysbiosis and will open new areas for researchers.

## 1. Introduction

The human body is home to billions of small living creatures known collectively as the human microbiota, and their genome is referred to as the microbiome. The gut microbiota, sometimes known as the “forgotten organ,” with roughly 3 million genes, which is up to 150 times the human genome [1]. Microbes flourish on our skin, as well as in our genitourinary, gastrointestinal, and respiratory systems, with the gastrointestinal tract being the most densely infested. The colon and rectum, located at the end of the gastrointestinal (GI) tract, are thought to house the greatest number of bacteria in the human body [2]. Surprisingly, just one-third of our gut microbiota is shared by most individuals, while the remaining two-thirds is unique to each person, similar to a personal identification card [3]. The makeup of this microbial population changes over time, and it is subject to both external and endogenous variations [4]. Diet, metabolism, age, location, stress, and antibiotic therapy may all disrupt the balance between helpful commensals and potentially pathogenic microorganisms, ‘Dysbiosis’ [5] is the word for this shattered equilibrium. The gut microbiota has been shown to have a crucial role in maintaining immune function and metabolic balance, vitamin generation, pathogen protection, stimulating angiogenesis, and maintaining the intestinal barrier. The realization that gut microbiota plays a role in maintaining homeostasis and regulating practically every major bodily system, including the central nervous system (CNS), has sparked a revolt in biomedicine during the last two decades [6]. The “gut–brain axis” (GBA) implies the existence of a two-way communication route between gut microorganisms and the CNS, is now widely accepted [7], and dysregulation of this axis is increasingly suspected of being involved in the pathophysiology of neurological disorders, such as Autism Spectrum Disorder, Alzheimer’s disease, multiple sclerosis, Parkinson’s disease, etc. [6]. Currently, microbiome-based therapies, such as prebiotics, probiotics, and synbiotics, as well as microbiota fecal transplants, aim to promote eubiosis to improve metabolic and mental health [8]. In addition, Boron (B), a kind of active bio-trace-element, has been suggested to be an essential nutrient, which imparts neuroprotective effects. Boron intake has been linked to bone, mineral, and lipid metabolism, and immunological function. As evidence mounts that B is critical for human health, it is critical to investigate probable links between B nutrient intake and brain and psychological function [9]. This review’s main goal is to summarize what is currently known about the gut microbiota’s function in different neurodevelopmental and neurodegenerative illnesses, as well as its changing makeup. Furthermore, the numerous therapy methods that have been employed to ameliorate these illnesses are discussed.

## 2. Gut Microbiota

### 2.1. Classification and Characterization

The gut microbiota is of bacteria classified by genus, family, order, and phylum. *Firmicutes* and *Bacteroidetes* are the most common gut microbial phyla, accounting for 90% to 95% of the total microbiota [10], which is also described as the genetic material of all bacteria in the intestine. *Clostridium* (95% of the *Firmicutes* phylum), *Blautia*, *Faecalibacterium*, *Enterococcus*, *Lactobacillus Eubacterium*, *Roseburium*, and *Ruminococcus* are among the more than 200 genera that make up the phylum *Firmicutes*. *Bacteroidetes* consist of predominant genera like *Bacteroides* and *Prevotella*. Other phyla in the gut microbiota include *Proteobacteria*, *Actinobacteria*, *Fusobacteria*, *Spirochaetes*, and *Verrucomicrobia* [11].

Researchers have been able to phylogenetically identify and quantify the components of the gut microbiota [12] using novel methods based on DNA extraction and sequencing of 16S rRNA, 18S, and ITS gene [13]. The 16S rRNA gene (1500 bp) is large enough for informatics purposes and is found in almost all bacteria, and its function has not changed over time thus it has been by far the most commonly used phylogenetic tool [14,15]. Metagenomics provide a wealth of information on a microbiome’s current taxonomies, but little about critical functions. Meta-transcriptomics is an RNA-based technique that studies the functional analysis of genes expressed by the microbiome and may also assess the taxonomic composition of the microbial population. Meta-proteomics is the study of the collective protein composition of multi-organism systems, and large-scale identification and quantification of proteins from microbial communities provide direct insight into the phenotypes of microorganisms on the molecular level [16]. Metabolomics is described as a systematic analysis of metabolites in a biological specimen that allows for detailed phenotyping of metabolic phenotypes and precision medicine on a number of levels, including the identification of metabolic derangements that underpin disease, the development of biomarkers that can be used to diagnose a disease or monitor therapeutic activity, as well as the discovery of new potential therapeutic [17].

### 2.2. Variation in the Composition of Gut Microbiota

The makeup of the intestinal flora is dynamic and is affected by a variety of parameters, including intestine anatomical area, gestational age, type of delivery, breastfeeding techniques, weaning duration, age, antibiotic usage, ethnicity, dietary habits, and cultural practices (Figure 1A) [18]. The complexity and quantity of bacteria tend to grow as we progress down the GI tract, with tiny numbers in the stomach but extremely high concentrations in the colon [19]. *Lactobacilli*, *Veillonella*, and *Helicobacter*, are the most common bacteria in the stomach, whilst *Bacilli*, *Streptococcaceae*, *Actinomycinaeae*, and *Corynebacteriaceae* are typical in the duodenum, jejunum, and ileum. The concentration of *Lachnospiraceae* and *Bacteroidetes* in the colon increases as we descend the ileum [19]. The gestational age at birth is a significant factor in gut microbial colonization. The gut microbiota of preterm newborns (born before 37 weeks) differs from that of term babies. Microbial colonization in preterm infants is characterized by limited microbial diversity, an increased number of potentially pathogenic and facultative anaerobic bacteria, such as *Enterobacter*, *Enterococcus*, *Escherichia*, and *Klebsiella,* and obligatory anaerobes, such as *Bifidobacterium*, *Bacteroides*, and *Atopobium* have lower levels [20]. Microbiota of infants born through vaginal delivery composed of *Bifidobacterium longum* and *Bifidobacterium catenulatum* and other facultative anaerobic species, such as *Escherichia coli*, *Staphylococcus*, *Bacteroides fragilis*, and *Streptococcus*. Infants born via C-section, on the other hand, have *Streptococci* as the dominant species resembling maternal skin microbiota and lower levels of the protective *Bifidobacterium longum* subspecies infantis (*B. infantis*) bacteria [21]. Breast-fed neonates have a more consistent and uniform population of microbiota than formula-fed infants, harboring two times more *Bifidobacterium* spp. and a lower number of *Clostridium* and *Streptococcus* species [22].

The human gut is not microbiologically sterile at birth; in actuality, bacterial colonization is a multidimensional procedure that starts in utero and is impacted by a variety of factors, such as mode of delivery, feeding method, etc. By the age of one year, a child’s microbiota composition has a characteristic abundance of *Akkermansia muciniphila*, *Bacteroides*, *Veillonella*, and *Clostridium botulinum* spp. In adulthood, the three bacterial phyla that dominate, i.e., *Firmicutes* (*Lachnospiraceae* and *Ruminococcaceae*), *Bacteroidetes* (*Bacteroidaceae*, *Prevotellaceae*, and *Rikenellaceae*), and *Actinobacteria* (*Bacteroidaceae*, *Prevotellaceae*, and *Rikenellaceae*), and as people become older, *Bifidobacterium* and *Firmicutes* levels tend to decline and *Bacteriodetes* and *Proteobacterium* levels ascent [12].

Dietary content has a significant impact on the gut microbial population [23]. High-fat diets are linked to lower numbers of Gram-negative and Gram-positive bacteria in the intestine, including *Bifidobacteria* [24]. Vegans showed lower levels of *Bacteroides*, *Bifidobacterium*, and *Enterobacteriaceae* spp. than control patients, according to a study by Zimmer et al. [25]. Wholegrain and prebiotic-rich diets have been demonstrated to lower opportunistic infections, including Enterobacteriaceae and Desulfovibrionaceae, while increasing Bifidobacteriaceae, which function as gut barrier-protecting bacilli [26]. Antibiotics are vital tools in the battle against infectious illnesses, but their usage can decline bacterial diversity and abundance, depending on the antibiotic class, dosage, and exposure time. Intrapartum antibiotics can increase the number of Bacteroides and Enterobacteria in the newborn gut while decreasing Bacteroidetes [27]. In addition, non-dietary lifestyle variables, such as lack of exercise or obesity, stress and smoking also lower the beneficial gut microbiota [28].

### 2.3. Important Functions and Beneficial Effects of Gut Microbiota in Our Body

The development, activation, and function of the host immune system are all influenced by the microbiota. The host immune system, in turn, has evolved several mechanisms for maintaining its symbiotic interaction with the microbiota [29].

The innate immune system, which is made up of physical and chemical barriers, immune cells, and blood proteins (e.g., cytokines), is our first line of defense. This differential detection of commensal and pathogenic bacteria is mediated by Toll-like receptors (TLRs) found on the membranes of epithelial and lymphoid cells in the small intestine. TLRs identify multiple microbe-associated molecular patterns (MAMPs), including diverse bacterial antigens, such as peptidoglycan components (muramic acid, flagellin, capsular polysaccharides, and lipopolysaccharides), and activate innate intestinal immunity. NOD-like receptors (NLRs) identify a variety of microbial-specific chemicals and cause the formation of inflammasomes, which can operate as damage-related pattern sensors. Paneth cells, which are specialized secretory cells of the small intestine mucosa, also play an important role in defining gut microbiota architecture by generating Antimicrobial Peptides (AMPs). Immune receptors are known as Pattern-Recognition Receptors (PRRs) play a dual role, detecting pathogens and symbionts, with differing outcomes for microorganisms and hosts [30].

Unlike innate immunity, adaptive immunity is highly specific to a given pathogen and is mediated by two primary lymphocyte populations known as B cells and T cells. The adaptive immune system relies heavily on CD4+ T cells. Naive CD4+ T cells can develop into one of four subtypes after being stimulated: T helper 1 (Th1), Th2, Th17, or regulatory T cells (Treg). Each lineage secretes a unique cytokine after differentiation [31]. *Bacteroides fragilis* induces a systemic Th1 response, which is critical for eradicating intracellular infections, through its polysaccharide amolecules. On the other hand, *Segmented Filamentous Bacteria* were shown to be effective inducers of Th17 cells, and Clostridia have been found to increase the production of colonic Tregs, an important mediator of immunological tolerance whose malfunction can lead to autoimmune diseases. The gut microbiome helps CD8+ T cells influence other peripheral immune cells, such as marginal zone B cells, plasmacytoid dendritic cells, and invariant natural killer T cells by training them. Though gut-associated B cells are present in Peyer’s patches, the gut microbiota is a significant driving force for mucosal IgA synthesis. Commensal bacteria and soluble antigens are coated by Secretory IgA (SIgA), which prevents them from attaching to the host epithelium and penetrating into the lamina propria. As a result, SIgA acts as an intestinal barrier and aids in the maintenance of a mutualistic relationship between the host and the microbiota [32].

They provide colonization resistance, i.e., protection from exogenous pathogen infection by competing for shared resources and habitats, as well as by strengthening host defense processes [33]. They support epithelial homeostasis because the intestinal epithelium is equipped with a variety of PRRs that identify microbial components and, as a result, activate cell signaling pathways that promote cellular growth [34]. The gut microbiome can also stimulate angiogenesis [35] and fat accumulation by activating local microvascular cells. They can also affect the nervous system, contributing to the development of naive microglia, with the number of mature microglia decreasing in the absence of microbiota. They also break down dietary molecules, synthesize vitamins and other nutrients, modify bone mass density, and metabolize medicines into active substances [36].

### 2.4. Microbiota and Its Metabolites

Various bacterial genera and species, in cooperation with the host, are responsible for the production of various metabolites. In the large intestine, indigestible oligosaccharides that escape digestion and absorption in the small intestine are fermented by intestinal microbiota producing short-chain fatty acids (SCFAs), such as acetate, propionate, and butyrate (>95%), with formate, valerate, caproate, or others accounting for the remainder [37]. Because most propionate is metabolized in the liver (Koh et al., 2016), acetate is the most common SCFA in the circulatory system, whereas butyrate is predominantly metabolized in the epithelial mucosa in the maintenance of colonic health (van der Beek et al., 2017). Firmicutes are the main producers of butyrate, whereas Bacteroidetes are the leading producers of acetate and propionate [38]. SCFAs are carried into host cells and interact with the G protein-coupled receptors GPR41, GPR43, and GPR109A, which sense metabolites and are expressed in the gut epithelium and immune cells. As a result, a process that is important for maintaining homeostasis in the intestine and other organs is induced [39]. The second mechanism is connected with inside-cell direct suppression of nuclear class I histone deacetylases (HDACs), such as HDAC1 and HDAC3. Inhibition of HDACs is mostly linked to anti-inflammatory immune phenotypes, such as decreased proinflammatory cytokine (IL-6, IL-8, TNF-α, and so on) and decreased NF-kB action. SCFAs can also maintain the epithelial barrier’s integrity by regulating tight junction proteins such claudin-1, occludin, and zonula occludens-1 (Wang et al., 2012). Thus, reduced levels of these proteins would aid bacteria and lipopolysaccharide (LPS) translocation, prompting an inflammatory response. SCFAs also play a role in colonocyte differentiation and proliferation, mucosal cell migration, mucin 2 expression augmentation, oxidative stress modulation, and immune response (van der Beek et al., 2017), all of which are crucial to the human body’s fight against diseases, such as Parkinson’s disease, Crohn’s disease, etc. [38]. In addition, SCFAs can be delivered to several organs after being metabolized, e.g., Propionate is mostly involved in gluconeogenesis, whereas acetate and butyrate are primarily involved in lipid biosynthesis (Figure 2) [38].

Tryptophan is a precursor for important metabolites and is a vital aromatic amino acid. In host cells, tryptophan from dietary sources, such as oats, fish, milk, and cheese can take one of two routes: kynurenine [40,41] or serotonin [42]. In the third route, gut microbes are involved in the direct conversion of tryptophan to a variety of compounds, including indole and its derivatives (Figure 3 and Figure 4) [43]. *Clostridium perfringens* and *Escherichia coli* generate indole derivates, such as indoleacetylglycine, indoxyl sulfate, indole-3-propionate, 6-sulfate, and serotonin which binds to the aryl hydrocarbon receptor, thus, help maintaining colon integrity by regulating the production of inflammatory and anti-inflammatory genes involved in the gut–brain axis [44,45]. Biogenic amines, such as trimethylamine-*N*-oxide (TMAO), trimethylamine, agmatine, histamine, etc., act on histamine receptors and are involved in gut epithelial homeostasis, cell growth, and aging; modulate anti-inflammatory and anti-tumoral effects. They are produced by *Clostridium saccharolyticum*, *Campylobacter jejuni*, *Bifidobacterium* [38]. Bile Acids (Bas) are cholesterol-derived small molecules produced by hepatocytes. The gut microbiota converts primary Bas (chenodeoxycholic acid and cholic acid) into secondary Bas and deconjugates them [43]. These Bas primarily target G protein-coupled bile acid receptor 1 and regulate glucose, cholesterol, and energy homeostasis, maintains intestinal barrier function, facilitate lipid-soluble vitamin absorption, and inhibit NF-kB-dependent transcription of proinflammatory genes. Vitamins, such as vitamin B9, thiamine, vitamin B2, niacin, vitamin K, vitamin B1, riboflavin, etc., are produced by bacteria, such as *Bifidobacterium bifidum*, *Bacillus Subtilis*, *Escherichia coli*, *Bacteroidetes,* and are involved in immunological function, cellular metabolism, cell proliferation and offer vitamin sources for hosts. Polyphenols produced by *Clostridium*, *E. coli*, *Salmonella*, and *Bacteroides* function as antioxidants, and reduce the risk of colon cancer and inflammation [38,46,47].

## 3. Gut–Brain Axis

The gut–brain axis (GBA) is characterized by bidirectional communication, i.e., the gut microbiota sends messages to the brain, and the brain sends signals to the gut via neurological (vagus nerve and enteric nervous system), endocrine (cortisol), immunological (cytokines), and humoral pathways [48]. The neuroendocrine and neuroimmune systems, which include enteroendocrine cells and gastrointestinal enterochromaffin cells, the intestinal mucosal barrier, and the blood–brain barrier, are primarily involved in bottom-up communication [49,50]. SCFAs, tryptophan metabolites, and secondary BAs, among other gut microbiota metabolites, are important mediators of this bottom-up communication. The neuroanatomical route, the modulation of the intestinal barrier, and the release of neurotransmitters (e.g., 5-HT and catecholamines) are all involved in top-down communication between the brain and gut bacteria [51,52]. Two neuroanatomical routes connect the gut and the brain. The first reflects direct communication between the gut and the brain via the vagus nerve in the spinal cord and the autonomic nervous system (ANS). The second is a bidirectional connection between the gut’s enteric nervous system and the spinal cord’s vagus nerve and ANS [53].

The hypothalamic–pituitary–adrenal (HPA) axis is a significant neuroendocrine system that regulates many physiological functions in response to psychological and physical stresses, such as infections, to ensure a proper reaction to the stressor [54]. The release of corticotropin-releasing hormone (CRH) from the paraventricular nucleus signals the activation of the HPA axis, which subsequently induces the release of adrenocorticotropic hormone (ACTH) from the anterior pituitary gland. ACTH stimulates the generation of glucocorticoids (cortisol in humans) in the adrenal cortex, which can have a significant influence on gut physiology (e.g., modifying the intestinal epithelial barrier and immunological responses) and gut microbiota composition [55]. Immune response activation, such as the production of chemokines and cytokines by immune cells (dendritic cells, B cells, mast cells, and T cells) in the gut or elsewhere, impact the brain. The gut microbiota produces a variety of important neurotransmitters, including gamma-aminobutyric acid, 5-HT, dopamine, and SCFAs, which have an impact on the human body, including the brain [53].

## 4. Role of Microbiota in Brain Development

The gut microbiota has been found to impact microglial maturity and function, blood–brain barrier formation and stability, myelination, and neurogenesis, along with other neurodevelopmental processes [56]. The relevance of the microbiome in early brain development has been discovered thanks to germ-free mice. Diaz Heijtz and colleagues (2011) discovered that germ-free mice have an upregulation of genes associated with a variety of plasticity and metabolic pathways, including long-term synaptic potentiation, steroid hormone metabolism, and cyclic adenosine 5phosphate-mediated signaling, with the cerebellum and hippocampus being the most affected, using a genome-wide transcriptomic approach [57]. Adult hippocampal neurogenesis is boosted in germ-free mice in the dorsal hippocampus (Ogbonnaya et al., 2015), but antibiotic treatment reduces neurogenesis, which can be restored with probiotics and exercise (Mohle et al., 2016). Increased turnover of dopaminergic and 5-HT neurotransmitters, as well as elevations in synaptogenesis markers, have been found in the striatum of germ-free mice, contributing to alterations in locomotive and exploratory behavior [58]. Luczynski et al., 2016 showed that germ-free mice have increased hippocampal and amygdala volume and have dendritic hypertrophy in the basolateral amygdala (mediate anxiety and fear-related response, as well as social behavioral patterns), when compared to mice with normal microbiota [59]. Germ-free mice had hypermyelination and elevated expression of genes involved in myelination in the prefrontal cortex part of the brain (Hoban et al., 2016) [57,60]. Lu et al. (2018) investigated the impact of preterm newborn microbiotas known to cause either high or low growth phenotypes on postnatal brain development utilizing a germ-free mouse transfaunation paradigm. When compared to the microbiome related to the high growth phenotype, the neuronal markers NeuN and neurofilament-L, as well as the myelination marker MBP, the microbiome associated with the low growth phenotype exhibited a drop in these markers. Furthermore, a poor growth phenotype-associated microbiota was linked to increased neuroinflammation, as shown by an increase in Nos1, as well as changes in the IGF-1 pathway, including lower circulating and brain IGF-1 and lower circulating IGFBP3 [61].

## 5. Gut Microbiota in Neurodevelopmental Diseases

The human brain begins to grow in the third week of pregnancy, and by the time it is born, there are 86 billion neurons and 100 trillion connections, creating basic circuits. Under the impact of the environment, these rudimentary circuits evolve into increasingly complex linked circuits. Hormones such as oxytocin, the immune system, neurotransmitters such as serotonin, and the microbiome-gut–brain axis all contribute to the neuronal circuitry that underpins social cognition, emotion, and behavior. These social, cognitive, and behavioral dimensions, as well as their neurodevelopment, are impaired in neurodevelopmental disorders, including ASD and schizophrenia. Even in those who are not predisposed to disease, the “pathogenic” microbiome may be enough to cause disease (Kim et al., 2017). Microbiota from individuals with depression, irritable bowel syndrome-associated anxiety, or schizophrenia, for example, was transplanted into wild-type mice and encouraged indication-specific behavioral abnormalities (De Palma et al., 2017; Zheng et al., 2016, 2019) [62,63].

### Autism Spectrum Disorder

Autism is a developmental disease marked by difficulty with social communication, as well as a lack of interest and repetitive conduct [64]. According to estimates from the CDC’s Autism and Developmental Disabilities Monitoring (ADDM) Network, the rate of autism has grown significantly across the globe (prevalence of 1%) [56], impacting 1 in 54 children [65]. However, determining the specific etiology and pathophysiology of ASD is challenging, and effective treatments are scarce. Genetic and environmental variables, aberrant immunological responses, and dysbiosis-induced gut integrity breakdown have all been related to the development of autism [66]. Prenatal nutrition, perinatal stress, cesarean birth, preterm, restricted breastfeeding, infections, and antibiotic usage are among the environmental risk factors linked to ASD [67,68]. Food rejection, food allergies, constipation–diarrhea, food intolerance, stomach discomfort, and fussy eating habits are all more frequent in children with ASD than they are in the general population [69]. Alterations in the GMBA have been linked to neurodevelopmental disorders, including ASD, according to current evidence [70,71]. Constipation was shown to be more common in children with ASD [72] and, might be linked to microbial dysbiosis, which could compromise the intestinal barrier’s integrity [73].

According to Srikantha et al., intestinal permeability produced by a loss in barrier-forming tight junctions might be a possible biomarker in ASD pathophysiology [74]. Several investigations have found that people with ASD had higher levels of IL-6, IL-1, TNF-α, and TGF-β in their serum, brain tissue, and spinal fluid [75]. Ashwood and colleagues supported the hypothesis that there is a high pro-inflammatory chemical circulation and a low regulatory circulation in patients with ASD, which further supports the occurrence of mucosal immunopathology. In comparison to controls, CD3+ T cells were observed to be more abundant in the duodenum and colon of children with ASD [76,77]. The significance of GMB metabolites, particularly short-chain fatty acids (SCFAs), in the pathophysiology of ASD has piqued researchers’ interest in recent years. These SCFAs are thought to affect the mitochondrial role in terms of the citric acid cycle and carnitine metabolism, as well as epigenetically altering ASD-related genes [78]. Butyrate (BT), one of the most prominent SCFAs, has been recommended as a neuroprotectant and has been shown to positively affect mitochondrial activity. By inhibiting HDAC, BT modulates the blood–brain barrier (BBB) and suppresses intestinal pro-inflammatory macrophage activity. As a result, BT-producing bacterial taxa are found in lower numbers in autistic people [79]. Propionate disrupts GI function in a way that causes problems in people with ASD (Getachew et al.) [80]. It has the potential to elicit reversible neuro-inflammatory, metabolic, behavioral, and epigenetic alterations similar to those seen in ASD animal models. According to Altieri et al., a high amount of urine p-cresol is linked to ASD-like repetitive behaviors [81]. In normal mice, 4-ethyl phenyl sulfate (4-EPS), a uremic toxin and a GMB metabolite, causes ASD-like behavior, as well as anxiety-like symptoms. Germ-free mice do not detect 4-EPS, showing that it is a GMB metabolite. The 4-EPS levels were found to be greater in the offspring of maternal immune activation (MIA), as indicated in a mouse model with ASD characteristics. In an MIA model of ASD, treatment with Bacteroides fragilis, a probiotic bacteria, lowered the quantity of 4-EPS [82]. Probiotics or prebiotics, as well as fecal microbiota transplant (FMT), have been studied in ASD clinical trials to improve GI abnormalities and ASD severity [83]. For a 19-week experimental period, 13 ASD children were given a probiotic (Vibosome containing Lactobacillus and Bifidobacterium), and it improved GI problems significantly [84].

In an open-label clinical experiment, Fecal Microbiota Transplantation from healthy controls to ASD-diagnosed children is performed to look for improvement in GI-related symptoms in ASD people. The study indicated that after seven to eight weeks of daily maintenance dosages, 80% of children with ASD had reduced GI symptoms and ASD severity [85]. A two-year follow-up of this cohort revealed that treated patients had improved their GI and ASD symptoms [86]. In comparison to probiotic and prebiotic therapies, FMT therapy exhibited a long-term benefit.

## 6. Microbiota in Neurodegenerative Disease

Through the GMBA, gut dysbiosis can influence brain immune homeostasis and play a role in the etiology of neurodegenerative illnesses, such as Parkinson’s disease (PD), Alzheimer’s disease (AD), multiple sclerosis, and amyotrophic lateral sclerosis [87].

### 6.1. Alzheimer’s Disease

Dementia is a condition in which memory, conduct, reasoning, capacity to do daily activities, judgment, and language deteriorate. Alzheimer’s disease and other forms of dementia have been claimed to be the fifth leading cause of mortality worldwide [88]. The most important risk factor is age, with the vast majority of people with Alzheimer’s dementia being 65 or older [89]. Extracellular -amyloid (A), senile plaques (SP), and intracellular neurofibrillary tangles (NFT) are the key characteristics of Alzheimer’s disease. Increased production of reactive oxygen species (ROS) causes neuroinflammation and cell death. In addition, vascular abnormalities and mitochondrial damage have a role in the etiology of Alzheimer’s disease [90,91].

#### 6.1.1. Gut Dysbiosis and Alzheimer’s Disease

The generation of signaling proteins that impact metabolic pathways relevant to AD development is affected by changes in the gut microbiota. The ageing process causes local systematic inflammation, which impairs GIT permeability and blood–brain barrier function, by modifying the GM composition, i.e., a higher abundance of pro-inflammatory bacteria than anti-inflammatory bacteria (Figure 5) [92].

#### 6.1.2. Metabolites Implicated in Alzheimer’s Disease

*Lipopolysaccharide (LPS)*—Lipopolysaccharide (LPS) is a lipid-sugar compound that is a prominent component of Gram-negative bacteria’s cell walls [88] (50–70% in the normal gut microbiota). LPS is a valuable tool for investigating neuroinflammation in neurodegenerative diseases [93]. Tight connections between intestinal epithelial cells prevented LPS from entering the bloodstream in healthy people. LPS will enter the bloodstream and produce inflammation if the tight connections are weakened. As a result, blood LPS levels indicate not only inflammation but also a leaky gut. A slew of in vivo and in vitro investigations have revealed that LPS activates many intracellular molecules that alter the expression of several inflammatory mediators, hence contributing to or initiating neurodegeneration. LPS activates TLR4-CD14/TLR2 receptors on leukocytes and microglia, resulting in NF-kB-mediated cytokine surges that raise Aβ levels, injure oligodendrocytes, and cause myelin damage in the AD brain. Because Aβ 1–42 is also a TLR4 agonist, it may set in motion a vicious loop that accounts for AD’s persistent progression [94]. The blood–brain barrier is also disrupted by serum LPS, that can also enter the brain and reactivate microglia, astrocytes, and numerous amyloidogenic and inflammatory pathways. Increased levels of inflammatory cytokines and NF-kB promote a rise in amyloid precursor protein (APP) and Aβ protein cleavage and accumulation, resulting in neuron loss and the development of Alzheimer’s disease [88]. Zhao et al. (2019) showed that LPS administration causes illness behavior and cognitive impairment, as well as microglia activation and neuronal cell death in the hippocampus in C57BL/6J mice. The LPS treatment decreased the levels of IL-4 and IL-10 while increasing the levels of TNF, IL-1, PGE2, and nitric oxide (NO). The NF-kB signaling pathway was activated in the LPS groups, according to Western blot analysis. In addition, VIPER, a TLR-4-specific inhibitory peptide, reduced LPS-induced neuroinflammation and cognitive impairment [95]. According to Thingore et al. (2020), LPS injection elevated neuroinflammation, caused poor memory retention and exacerbated the cognitive decline, and led to oxidative stress by lowering SOD, and increasing lipid peroxidation [96].

*Amyloid*—Amyloids are self-aggregating proteins that can induce cellular dysfunction in patients with neurodegenerative disorders [97]. Aβ is a cleavage product of APP, a transmembrane protein implicated in neuronal growth, signaling, and intracellular transport [98]. GM-produced amyloids have been shown to cross-seed Aβ deposition in several in vitro and in vivo studies [99]. Curli is created by Escherichia coli, TasA is made by Bacillus subtilis, CsgA is produced by Salmonella Typhimurium, FapCP is produced by pseudomonas fluorescens, and so on [100]. Bacterial amyloids have a different basic structure than brain amyloids, although they have similar metabolic and structural properties [101]. In a process known as seeding, preexisting amyloid aggregates produced from the same protein can speed up the polymerization of amyloidogenic proteins into ordered fibers. These amyloids cause Aβ fibrils and oligomers to misfold, allowing bacteria to attach to one another and create biofilms that may withstand immunological or physical attack. Bacterial amyloid proteins in the gut may prime the immune system, increasing immunological responses to intrinsic neural amyloid formation in the CNS [102]. Resemblances in tertiary protein structure may play a role in the development of prion-like agents via molecular mimicry, which results in cross-seeding, in which an amyloidogenic protein induces the production of another protein, such as a host protein with a distinct structure, to adopt the pathogenic sheet structure. According to Cattaneo et al. (2017), amyloidosis-positive individuals had greater blood levels of IL-1β, IL-6, C-X-C motif chemokine ligand, and nod-like receptor protein 3, and lower levels of anti-inflammatory cytokine IL-10 [103]. Ho et al. (2018) found that the gut microbiota can help guard against Alzheimer’s disease by promoting the production of certain SCFAs that prevent the creation of harmful soluble Aß aggregates [104]. In recent work, Javed et al. (2020) found that FapCS has a catalytic ability in seeding peptide amyloidosis, poor cognitive function, and behavior pathology in vitro, in silico, and in a zebrafish AD model [105].

*Calprotectin* is a tiny calcium-binding protein generated by neutrophils and monocytes, a heterodimer of S100A8/A9 (a TLR4 ligand). Elevated fecal calprotectin may act as a sign of intestinal inflammation. The concentration of fecal calprotectin in 22 individuals with Alzheimer’s disease was compared to serum amounts of aromatic amino acids by Leblhuber et al. (2015). Increased fecal calprotectin concentrations are linked to impaired intestinal barrier function in Alzheimer’s patients [106].

#### 6.1.3. Leaky Gut and Leaky Brain

The mucus layer, intestinal epithelium, and lamina propria [107] form the intestinal barrier, protecting the body from pathogenic germs and preventing toxic particles, chemicals, bacteria, and other health-threatening organisms from entering the bloodstream. The makeup of the microbiota influences the permeability of the mucus layer [108]. The multitude of mucin-degrading bacteria Akkermansia muciniphila improves the gut barrier function and systemic inflammation [109]. Changes in tight junctions are mediated by pathogenic *E. coli* strains, Salmonella, Shigella, Helicobacter pylori, Vibrio, or Clostridium [110]. Increased intestinal permeability, often known as leaky gut, is caused by problems with the tight junctions’ competence. By cleaving E-cadherin (a cell adhesion molecule), Bacteroides fragilis exotoxin disrupts adherence junctions [111]. Disruption in gut homeostasis negatively impacts gut permeability by lowering beneficial substances, such as SCFAs and H2, and increasing harmful substances, such as LPS, amyloids, and TMAO, making the intestinal mucosal barrier permeable, activating peripheral immune responses, and raising peripheral and central Oxidative Stress levels [112]. The BBB (blood–brain barrier), which is made up of specialized brain endothelial cells, astrocytes, and pericytes, is a highly selective semipermeable boundary [113]. The BBB integrity is critical for brain growth and function. According to recent research, a variety of chemicals can compromise the BBB, allowing molecules, such as protein, viruses, and even bacteria to enter the brain and endanger brain health (Welling et al., 2015) (Table 1). The BBB’s structural and functional breakdown may be an early and crucial phase in the etiology of Alzheimer’s disease [114]. Pro-inflammatory and cytotoxic events result from a deposition in the vasculature, contributing to increased BBB permeability in the AD brain (Roher et al., 2003, Carrano et al., 2011, Erickson and Banks, 2013). TJs are disrupted by Aβ1-42 oligomers, which suppress the expression of ZO-1, claudin-5, and occludin while promoting the production of matrix metalloproteases (MMP)-2 and MMP-9. It also binds to the RAGE receptor and causes the formation of ROS, which disrupts TJs and compromises BBB integrity (Carrano et al., 2012). Tau may also induce BBB degeneration, according to in vitro studies and transgenic mice tauopathy models. Both tau and Aβ may, thus, contribute to the breakdown of the BBB, exacerbating the neurodegenerative process and the inflammatory reactions that accompany it [114].

### 6.2. Parkinson’s Disease

Parkinson’s disease (PD) is the world’s second most prevalent neurodegenerative illness, characterized by an aberrant buildup of α-synuclein fibrils known as Lewy bodies (LBs) in dopaminergic neurons in the substantia nigra (SN) [129]. It has a global incidence of 10–50 per 100,000 people per year and a prevalence of 100–300 per 100,000 people, with the number of persons with PD anticipated to double by 2030 owing to global population aging [130]. Increased intestinal permeability and systemic exposure of bacterial endotoxins are caused by changes in the gut microbiota, which causes excess α-syn expression and supports its misfolding to generate LBs. The intestinal LBs will enter the CNS via the vagal nerve and eventually travel to and destroy the substantia nigra [131], resulting in the formation of clinical signs of Parkinson’s disease, such as tremors, stiffness, balance issues, and loss of spontaneous movement (akinesia). Constipation is the most prevalent premotor sign in Parkinson’s disease, involving more than 70% of individuals and advancing pathogenesis more than 10 years before clinical symptoms appear. As a result, the symptom of constipation is considered a clinical biomarker for identifying prodromal PD (Berg et al., 2015) [132]. In individuals with PD, there was a significant drop in numerous gut microbiota metabolic products, which might lead to constipation. When intestinal infection was present, a higher vulnerability to PD was reported, which might trigger PD-like symptoms. In a mouse model, PD-derived gut microbiota might exacerbate α-synuclein-mediated motor impairments and brain disease, whereas germ-free mice displayed milder α-synuclein pathology (Sampson et al., 2016) [133]. The microbiome-related changes in PD are discussed in Table 1.

### 6.3. Multiple Sclerosis (MS)

MS is a chronic autoimmune illness in which immune cells target the myelin sheath, causing demyelination and axonal loss, which leads to paralysis since myelin permits electric impulses to flow through neurons [134]. Despite multiple risk variables implicated in the development of autoimmune diseases, the gut microbiome is thought to be the most important environmental risk factor for MS [135]. MS patients had a lower number of Faecalibacterium, Eubacterium rectale, Corynebacterium, and Fusobacteria, and a higher proportion of Escherichia, Shigella, Clostridium, and Firmicutes compared to healthy controls [136,137]. The most extensively used animal model that matches the characteristics of MS in humans is EAE (Experimental Autoimmune Encephalomyelitis). EAE is not induced in GF mice, suggesting that the gut microbiota is essential for EAE induction. Oral therapy with ampicillin, vancomycin, neomycin, sulfate, and metronidazole produced a similar response, with a delay in the beginning and reduction in the severity of the illness, as well as lower levels of pro-inflammatory cytokines and higher levels of interleukin IL-10 and IL-13 [138]. Lipid 654 is expressed in considerably reduced quantities in the blood of MS patients compared to both healthy persons and those with Alzheimer’s disease, according to Farrokhi et al. (2013) [139]. Probiotics (IRT5 including Lactobacillus casei, Lactobacillus acidophilus, Lactobacillus reuteni, Bifidobacterium bifidum, and Streptococcus thermophilus) were given before the induction of EAE, which led to a delayed start and milder duration of the disease [140]. Dysbiosis in MS is further described in Table 1.

### 6.4. Amyotrophic Lateral Sclerosis (ALS)

ALS is a deadly neurodegenerative disease that affects the neurons of the brain and spinal cord, resulting in the premature death of motor neurons [141]. Because of respiratory paralysis, the majority of ALS patients die within 3 to 5 years [142]. A number of studies have discovered indications of abnormalities in the gut microbiota in people with amyotrophic lateral sclerosis. Using an ALS animal model, Wu et al. (2015) discovered that tight junction structure was disrupted and intestinal permeability was enhanced. Gut dysbiosis has also been seen in ALS mice, with lower numbers of butyrate-producing bacteria, such as *Butyrivibrio fibrisolvens* and *E. coli* [143]. Fang et al. (2016) discovered a decreased Firmicutes/Bacteroidetes ratio, a large reduction in the genera *Anaerostipes*, *Oscillibacter*, and *Lachnospiraceae* (beneficial bacteria), and a significant rise in glucose metabolizing Dorea in ALS patients [144]. Using an ALS mouse model and a diet supplemented with 2% butyrate in drinking water, Zhang et al. found that intestinal microbial equilibrium was restored, gut integrity was enhanced, and life duration was extended as compared to control mice [145]. Further studies related to microbiota and ALS are discussed in Table 1.

## 7. Effect of Prebiotic, Probiotic, Synbiotic, and Psychobiotic Supplementation on Gut Microbiota and Associated Disorders

Prebiotics are substrates that are “selectively used by host bacteria, imparting a health advantage”. They are made up of non-digestible fibers, such as oligosaccharides, that operate as a particular substrate for probiotics in the GI tract, promoting growth and improving function [146]. Probiotics are “live bacteria that bestow a health benefit on the host when supplied in suitable amounts” [147]. Synbiotic refers to a probiotic and prebiotic combination. Probiotics and prebiotics have been shown to have a beneficial impact in the prevention of Alzheimer’s disease, Parkinson’s disease, depression, autism spectrum disorders, and other neurological and mental diseases [148]. The terminology “psychobiotics” was invented to explain probiotics and/or prebiotics therapeutic effects on mental health through immunological, humoral, neuronal, and metabolic pathways. [149]. “The excretion of acids (lactate, acetate), competition for nutrients and gastrointestinal receptor sites, immunomodulation, and the creation of specialized antimicrobial agents” are among the mechanisms of victorious probiotics [150]. Tsilingiri et al. define postbiotics as “any molecule released by or created through the metabolic activity of the microbe that has a favorable impact on the host, either directly or indirectly” [151]. The effect of various prebiotic, probiotic, and synbiotic formulations on neurological disorders in animal models can be seen in Table 2. Despite this wealth of knowledge, the real benefits of probiotics and prebiotics are largely unknown, and when research is compared, there are several gaps and disparities. Human studies are needed to improve the composition of the flora in specific patient groups, as well as the efficacy and safety of probiotics and prebiotics [148].

## 8. Boron as a Neuroprotective Agent

Boron (atomic number 5) is a nonmetallic solid member of group 13 of the periodic table. A vital mineral may be found in both food and the environment. Boron is essential for the activity of several metabolic enzymes, as well as the metabolism of steroid hormones and a variety of micronutrients, such as calcium, magnesium, and vitamin D [164]. In addition, it is important for the growth and maintenance of bone, reduction in inflammatory biomarkers, such as high-sensitivity C-reactive protein (hs-CRP) and tumor necrosis factor α (TNF-α), increases the levels of antioxidant enzymes, such as superoxide dismutase (SOD), catalase, and glutathione peroxidase, enhances electrical activity in the brain, cognitive performance, and short-term memory in the elderly, is effective in preventing and treating malignancies, such as prostate, cervix, and lung cancers [165]. Since 2003, bortezomib (Velcade), a proteasome inhibitor with in vitro and in vivo action against a range of tumors, has been utilized in clinical trials to treat malignant cancers (Figure 6) [166].

The approval of Velcade as a proteasome inhibitor causes interest in boronic acids in medicinal chemistry, which further lead to the discovery of two other drugs ixazomib and vaborbactam. Ixazomib approved by FDA in 2015 is used for the treatment of multiple myeloma. Vaborbactam approved by FDA in 2017 is a β-lactamase inhibitor, and has been used in combination with antibiotics for the treatment of urinary infections. Benzoxaborole containing drugs, Tavaborole and Crisaborole, has been approved by FDA in 2014 and 2017 and are used for the treatment of onychomycosis and Eczema, respectively [167,168,169].

Some boron-containing substances have also been reported to inhibit the phosphodiesterase 4 enzyme (PDE4) and inflammation-related cytokine release, both of which have been linked to improved cognition in aging and Alzheimer’s disease [170].

**Boron in Diet**—In 1904, Wiley found that consuming more than 500 mg/day (77 mg boron per day) of boric acid for 50 days caused abnormalities in appetite, digestion, and health, and he concluded that 4000 mg/day (699 mg boron per day) was the maximum beyond which a normal man could not go without injury. Following his research, the notion that boron represented a health danger gained traction [171]. Many countries throughout the world began prohibiting the addition of borates to food by the mid-1920s, although these prohibitions were loosened during World War II. Restrictions were gradually re-imposed after the war. The belief that boron had little nutritional value in higher animals or humans began to alter in the early 1980s. This was the year when a study found that boron deficiency increased gross bone deformities in chicks given low doses of vitamin D [172]. Boron is a mineral that can be found in leafy green foods, such as kale and spinach. Grains, prunes, raisins, noncitrus fruits, and nuts all contain it. The average person’s daily diet contains 1.5 to 3 milligrams (mg) of boron. Apples, coffee, dry beans, milk, and potatoes are the five most prevalent sources of boron in a person’s daily diet. Boron intakes of 1–3 mg/day have been shown to improve bone and brain health in adults when compared to intakes of 0.25–0.50 mg/day [173]. The right amount of boron can help to create the intestinal organizational structure, which improves gastrointestinal absorption [173].

Adult frog males that were deficient in boron had atrophied testes, lower sperm counts, and sperm dysmorphology. Female frogs had atrophied ovaries, and oocyte maturation was hindered. Boron deficiency resulted in a significant rise in necrotic eggs, as well as a high incidence of aberrant gastrulation characterized by yolk hemorrhage and exogastrulation [174].

**Boron and Microbiome**—Evariste et al. used boron nitride nanotubes (BNNT), which comprise hexagonal-boron nitride and boron, in their experiment. Multiple endpoints in the tadpoles, as well as bacterial populations associated with the host intestine were measured after the exposure. BNNT exposure boosted the tadpoles’ growth and it was also linked to gut microbiome remodeling, with taxa from the phylum Bacteroidetes benefiting. The findings support the conclusion that BNNT are biocompatible, as evidenced by the absence of harmful effects from the nanomaterials studied [175].

According to the findings of Wang et al. [176] the superabsorbent resin with boron (SARB) can boost bacterial community diversity in maize straw. Proteobacteria were found responsible for the absolute advantage of the bacterial population in the peat substrate and maize straw groups in 10 treatments. The superabsorbent resin with boron (SARB) synthesized in the laboratory, on the other hand, cannot change the original structure of the bacterial community and has a little toxic effect on the bacterial community in both peat substrate and maize straw, and, has an enhancing effect on Proteobacteria and Actinobacteria and a waning effect on Acidobacteria and Firmicutes to some extent [176]. These examples [175,176] clearly demonstrate the role of boron in modulating microbiome physiology.

**Boron as an anti-inflammatory agent**—Boron has been shown to lower the levels of inflammatory biomarkers in several studies. A considerable increase in plasma boron concentrations occurred 6 h following supplementation with 11.6 mg of boron, together with significant decreases in hs-CRP and TNF-α levels, in a recent human experiment, including healthy male volunteers. One week of 10 mg/d boron supplementation resulted in a 20% reduction in TNF-α, from 12.32 to 9.97 pg/mL, as well as significant reductions (nearly 50%) in plasma concentrations of hs-CRP, from 1460 to 795 ng/mL, and IL-6, from 1.55 to 0.87 pg/mL [177]. Boron reduces the synthesis and activity of serine protease enzymes implicated in the inflammatory response, according to animal experiments in which rats with induced arthritis benefited from orally or intraperitoneally supplied boron [178]. Calcium fructoborate a naturally occurring, plant-based boron-carbohydrate complex—has been demonstrated to have anti-inflammatory effects on cellular cultures, an open-label pilot research indicated that calcium fructoborate had highly beneficial effects on Osteoarthritis (OA) symptoms [179]. Scorei and colleagues then conducted a double-blind study in middle-aged patients with primary OA to see how different doses of calcium fructoborate affected systemic inflammation and dyslipidemia indicators. They found that all groups except the placebo group saw a reduction in inflammatory biomarkers CRP, fibrinogen (FBR), and erythrocyte sedimentation rate (ESR) after 15 days of food supplementation with calcium fructoborate [180].

**Boron in Neurodegenerative Disorders**—Nutritional intakes of boron have been shown to have favorable benefits on central neurological function, but the evidence is less conclusive than in the case of bone. They are, nevertheless, among the most receptive to the idea that boron is a healthy bioactive element for humans. Even though boron compounds have strong immunomodulatory properties, only a few research have looked at their potential to cure neurological illnesses like Alzheimer’s or Parkinson’s disease.

As we have already discussed, Alzheimer’s disease (AD) is characterized by amyloid (Aβ) aggregation, hyperphosphorylated tau, neuroinflammation, and memory impairment. Using in vitro and in vivo models of Alzheimer’s disease, Maiti et al. compared the therapeutic effects of trans-2-phenyl-vinyl-boronic-acid-MIDA-ester (TPVA) (**1**) and trans-beta-styryl-boronic-acid (TBSA) (**2**) (Figure 7). They found that TBSA prevented Aβ42 aggregation and enhanced cell survival more efficiently than TPVA. The benefits of TBSA were extended to C. elegans expressing Aβ42 and the 5xFAD animal model of Alzheimer’s disease and it was seen that TBSA prevented recognition- and spatial-memory deficits, and reduced the number of pyknotic and degenerative cells, GFAP levels, and Aβ plaques [181].

The findings of Maiti et al. (2020) back up those of Penland, who discovered that dietary boron intake significantly improves brain function and cognitive functioning in humans. Similarly, electroencephalograms showed that boron pharmacological intervention after boron deficiency improved functioning in older men and women, such as less drowsiness and mental alertness, better psychomotor skills (for example, motor speed and dexterity), and better cognitive processing (e.g., attention and short-term memory). In a separate study, Nielsen and Penland found that boron deficiency in rats reduced the number, distance, and time of horizontal movements, front entry, margin distance, and vertical breaks and jumps in spontaneous activity evaluations when compared to rats given boron supplementation [182]. These findings back up prior study, indicating that boron compounds can help with both impaired recognition and spatial memory problems.

Lu and colleagues (2012) identified a series of boron-containing compounds that act as Aβ aggregation inhibitors (**3**), antioxidants, and metal chelators in the treatment of Alzheimer’s disease [183]. Curcumin is currently being studied for its potential to treat a variety of cancers, as well as to prevent neuronal damage in Alzheimer’s disease. However, due to its low stability and solubility in aqueous solutions, its clinical utility is limited. Thus, Azzi et al. proposed a completely new class of boronated monocarbonyl analogues of Curcumin (**4**) (BMAC), in which a carbonyl group replaced the Curcumin β-diketone activity, and one of the two phenolic rings is replaced by an ortho-carborane (an icosahedral boron cluster). Furthermore, the effectiveness of BMAC (**4**) in inhibiting the development of amyloid aggregates was tested, and it was discovered that a compound that includes two OH moieties, outperforms Curcumin. The presence of a second -OH group can improve the binding efficiency of the chemical with β-amyloid aggregates [182].

Studies have also shown that CRANAD-28 (**5**), a difluoroboron curcumin derivative, may successfully identify amyloid-beta plaques for imaging both ex vivo and in vivo. The imaging brightness of CRANAD-28 (**5**), as well as its ability to pass the blood–brain barrier and low toxicity, making it a potentially useful imaging tool in Alzheimer’s research [183]. Given Alzheimer’s disease’s complex nature and pathophysiology, treatment techniques are being developed to integrate the benefits of each single-target therapy into a single molecule. Ritacca et al., using density functional theory, investigated the antioxidant property of boron as a radical scavenger and metal chelator. In aqueous and lipid settings, the most feasible radical scavenger mechanisms, hydrogen transfer, radical adduct formation, and single-electron transfer, were thoroughly investigated. The ability of metal chelation was explored by looking at the complexation of Cu(II) ion, one of the metals that can even catalyze the amyloid aggregation in excess [184].

Sorout and colleagues discovered that BN nanoparticles of various curvatures hinder the peptide’s conformational transition to its β-sheet form, that plays a key role in the aggregation and subsequent fibrillization of amyloid. This inhibition of β-sheet formation is by stabilizing the helical structure of the peptide (BNNT with the highest surface curvature), by making more favorable pathways available for transitions of the peptide to conformations of random coils and turns (planar BNNS), showing that BN nanoparticles have the potential to act as effective tools to prevent amyloid formation from Aβ peptides [185]. Novel boron-based compounds (BBCs) have been produced and tested as prospective candidates for the development of new Alzheimer’s disease treatments (AD). Cacciatore et al. demonstrated that a novel boron-based hybrid containing an antioxidant portion, i.e., BLA (**6**), inhibited cell death induced by Aβ1-42-exposure, increased cell viability, counteracted oxidative status, and inhibited acetylcholinesterase (AChE) (22.96% at 50 M), an enzyme whose enzymatic activity is increased in Alzheimer’s patients. These findings suggest that boron-based hybrids could be used to produce new medications for the treatment of Alzheimer’s disease [186].

Jiménez-Aligaga and co-workers described a series of boronic acid and boronate esters targeting pathways involved in pathogenesis of Alzheimer’s diseases. Compound **7** was found to be more active and responsible for nerve cell dysfunction in neurological disorders. Moreover, compound **7** enhanced the glutathione/glutathione disulfide (GSH/GSSG) ratio due to an increased GSH level (32 nmol/ mg protein), which is important for neuroprotection against ROS production, with an IC_50_ of 2.85 μm [187].

Jung and team demonstrated chalcone-derived boronic acid fluorescent probes for the detection of β-amyloid plaques in AD. Compound **8** displayed considerable enhancement in fluorescence in the synthetic β-amyloid aggregates for their fluorescence responses. Compound **8** efficiently binds to β-amyloid aggregates (KD = 0.79 ± 0.05 μm) due to the presence of boronic acid and strongly stained the β-amyloid in the experimental mice. Therefore, this study described boron-based compound could be a potential sensor to study neuronal functions in AD [188].

ALS is a neurological disorder caused by a mutation in human gene that encodes for secretory RNase angiogenin (ANG). A synthetic BA mask was reported by Hoang and coworkers, that inhibits ribonucleolytic activity of ANG thus imparting neuroprotective effect. Under normal physiological settings, compound **9** is inactive, but in the presence H_2_O_2_ the boron–carbon link in PBA is oxidatively cleaved, releasing the active ANG selectively in cells suffering from ROS-mediated damage [189].

In an experimental Parkinson’s disease model, Kucukdogru and colleagues found that boron nitride nanoparticles protect neurons from MPP+ (1-methyl-4-phenylpyridinium)-induced apoptosis. The use of hBNs, or hexagonal boron nitride nanoparticles, enhanced cell survival in the PD model as compared to MPP+ treatment. Furthermore, after using hBNs, antioxidant capacity increased while oxidant levels decreased. Finally, the findings showed that hBNs have a great deal of potential against MPP+ toxicity and could be employed as a novel neuroprotective agent and drug delivery method in the treatment of Parkinson’s disease [190]. These findings back the theory that boron exerts neuroprotective effects and is crucial for preserving the memory function.


**Other Neuroprotective Agents**


Apart from boron other neuroprotective agents are available in literature; some are briefly discussed here. Curcumin is an anti-bacterial, anti-oxidant, anti-inflammatory, and anti-tumoral polyphenol. Curcumin promotes beneficial bacterial strains, improves intestinal barrier function, and reduces the development of pro-inflammatory mediators. Curcumin therapy improved spatial learning and memory in APP/PS1 mice, a model of Alzheimer’s disease, suggesting its neuroprotective qualities [191,192]. Glutamate and its receptors are important in synaptic plasticity, mechanism that underpins learning and memory [193]. Therefore, disruption in their normal signaling plays a key role in a variety of neuropathological disorders, including Alzheimer’s disease, Parkinson’s disease, and schizophrenia, making them promising treatment targets [194]. Gut bacteria with glutamate racemase, such as *Corynebacterium glutamicum*, *Brevibacterium lactofermentum*, and *Brevibacterium avium*, can convert l-glutamate to d-glutamate, influencing the glutamate NMDAR and perhaps improving cognition in Alzheimer’s and Parkinson’s disease patients. As a result, gut microbiota and glutamate could be used to develop innovative dementia treatments [195].

Statin medication has been shown to alter the composition of the gut flora in recent research. In HFD-fed C57BL/6 mice, rosuvastatin altered gut microbiota and greatly increased the number of the family Lachnospiraceae, as well as the genera Rikenella and Coprococcus [196]. Atorvastatin reduces the microglia-mediated neuroinflammation, promotes intestinal barrier function by increasing protein levels of occludin and mucoprotein 2 and regulates the intestinal immune function by decreasing MCP-1, TNF-α, and increasing IL-10. Furthermore, atorvastatin alters the microbial composition by elevating Firmicutes and Lactobacillus and decreasing Bacteroidetes. It also decreases the amount of circulating endotoxin such as lipopolysaccharide-binding protein, a biomarker for leaky gut [197].

## 9. Conclusions and Future Perspective

The microbiome has established itself as an important component of the gut–brain axis, as well as a cornerstone in both health and illness. The gut–brain axis is a frontier of area of research, and several studies have found that changes in gut microbiota composition play a key role in the etiology of neurological illnesses, such as ASD, PD, AD, MS, and ALS. Dysbiosis can promote neuroinflammation by increasing inflammatory cytokines and bacterial metabolites, which can change gut and BBB permeability. The use of therapeutic substances, such as prebiotics and probiotics, to alter the gut microbiota opens up a potentially promising technique for treating a variety of neurological illnesses. The use of probiotics, prebiotics, and fecal microbiota transplantation resulting from breakthroughs in gut microbiome research to restore the dysbiosis-associated illness state’s, offers considerable potential as an alternative treatment approach in a variety of symptomatic disease management. Accurate identification of critical microbiota members, complex selection of microbial strains employed in probiotics, or different forms of prebiotics provided to selectively enumeration the ideal commensal have all added to the hurdles of applying microbiome-based therapy to clinical practice in the future. Possible applications of microbiome-based disease diagnosis, prognosis monitoring, prevention, and treatments, which have the potential to revolutionize present disease management and treatment methods, are certainly worth looking forward to. One area of medication research that is garnering increased attention is the use of boron compounds, which are being recommended as prospective treatments for lowering neuroinflammation and cognitive deficits [198,199]. In addition, boron-based diet and boron chemicals will play a significant role to improve dysbiosis and will open new windows for researchers in coming years. In this quickly expanding field of study, it is likely that we’ve barely scratched the surface, and the multiple techniques that have gotten us thus far will be visible in tackling the fascinating challenges that lie frontwards.

## Figures and Tables

**Figure 1 molecules-27-03402-f001:**
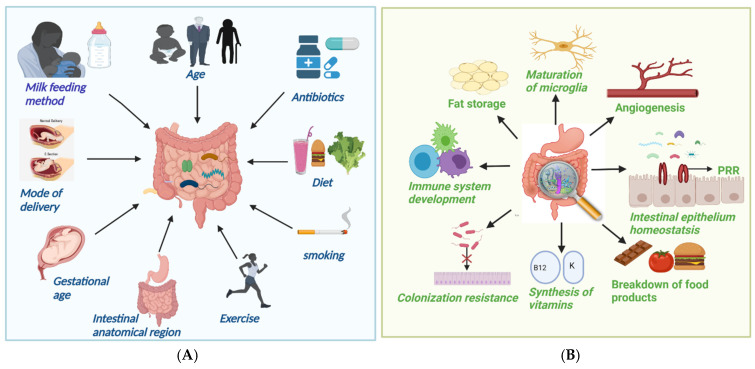
(**A**) Factors causing alteration in gut microbiota; (**B**) Important function of gut microbiota.

**Figure 2 molecules-27-03402-f002:**
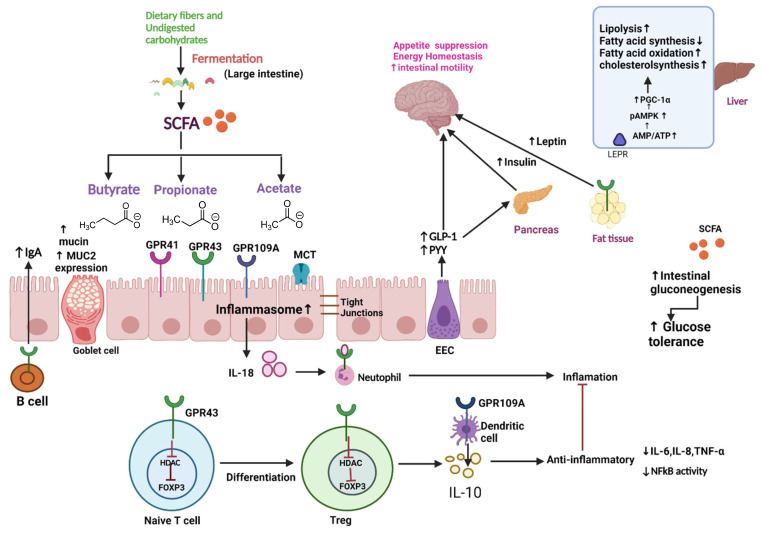
**Schematic overview of microbiota and its metabolites**—In the large intestine, dietary fiber and undigested carbohydrates are fermented into SCFA (Butyrate, propionate, and acetate) by gut microbiota. These metabolites can regulate tight junction, increase mucin production, upregulate MUC-2 expression, and increase synthesis of IgA through B cell activation. These metabolites also exert an anti-inflammatory effect by inhibiting HDAC (which leads to the differentiation of naïve T cells to regulatory T cells), which further decreases the production of proinflammatory cytokines, such as IL-6, IL-8, TNF-α, and NFkB activity. SCFA also stimulates intestinal gluconeogenesis, improving glucose tolerance. Moreover, SCFA can stimulate EEC to release GLP-1 and PYY, which acts as an anorexigenic agent by suppressing appetite and GLP-1 stimulate the pancreas to release insulin, thus increasing the uptake of glucose in muscle and adipose tissue. SCFA suppresses appetite by increasing leptin synthesis in adipose tissue. In liver SCFA, phosphorylate and activate AMPK directly by increasing the AMP/ATP ratio or indirectly via LEPR. Activation of AMPK triggers PGC-1α expression, which promotes fatty acid oxidation, lipolysis, and cholesterol synthesis and decreases fatty acid synthesis. Abbreviations: SCFA, Short-Chain Fatty Acid; MUC2, Mucin2; HDAc, Histone acetylation; EEC, Enteroendocrine cell; GLP-1, Glucagon-like Peptide 1; PYY, Peptide YY; pAMPK, phosphorylated Adenosine monophosphate activated protein kinase; AMP, adenosine monophosphate; ATP, adenosine triphosphate; LEPR, leptin receptor; PGC-1α peroxisome proliferator-activated receptor gamma coactivator 1-alpha.

**Figure 3 molecules-27-03402-f003:**
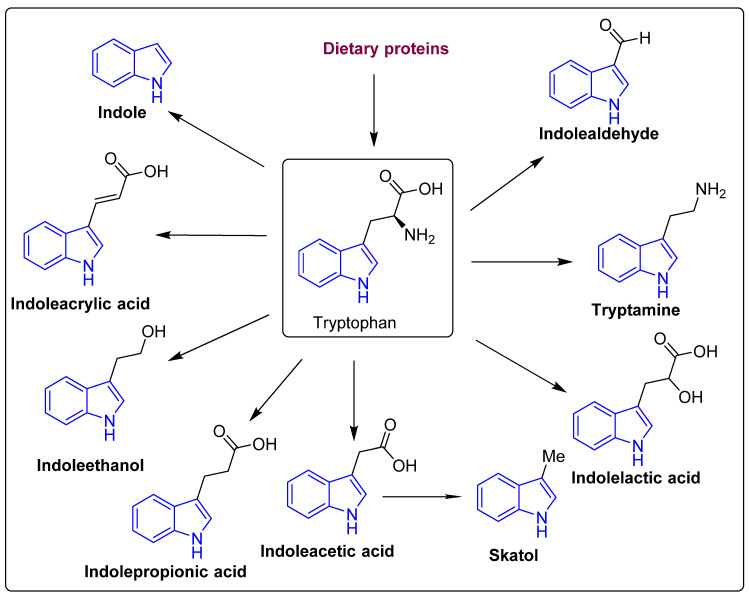
The structure of the tryptophan catabolites, indole, and its derivatives.

**Figure 4 molecules-27-03402-f004:**
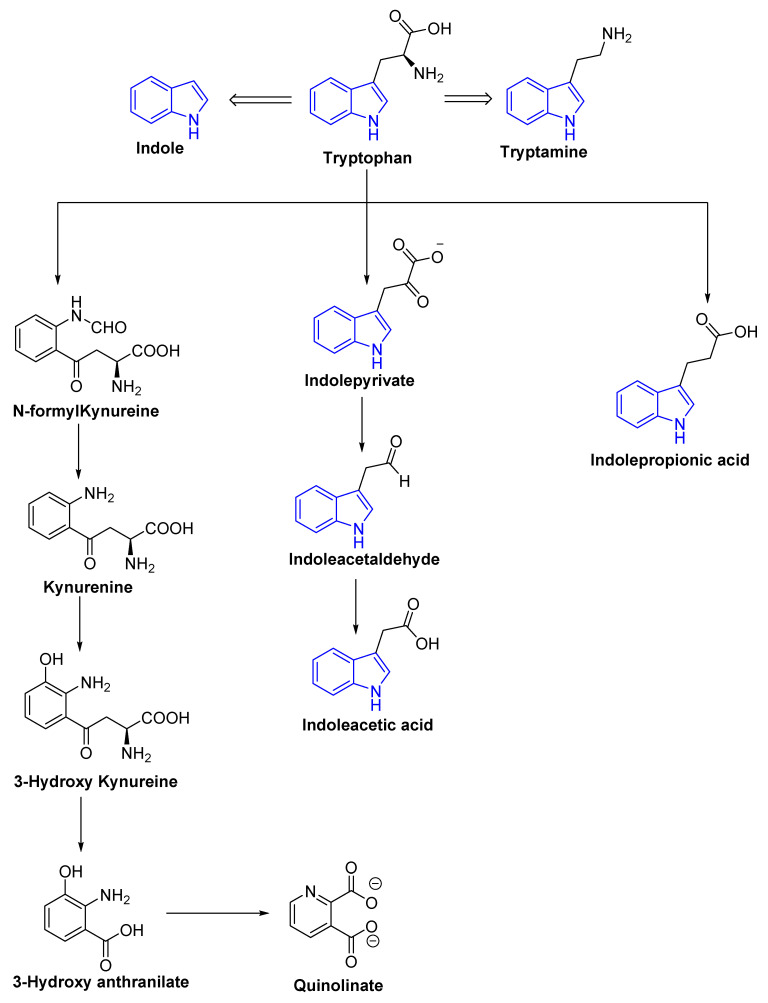
Schematic representation of tryptophan metabolism pathway leading to the production of neuroactive compounds.

**Figure 5 molecules-27-03402-f005:**
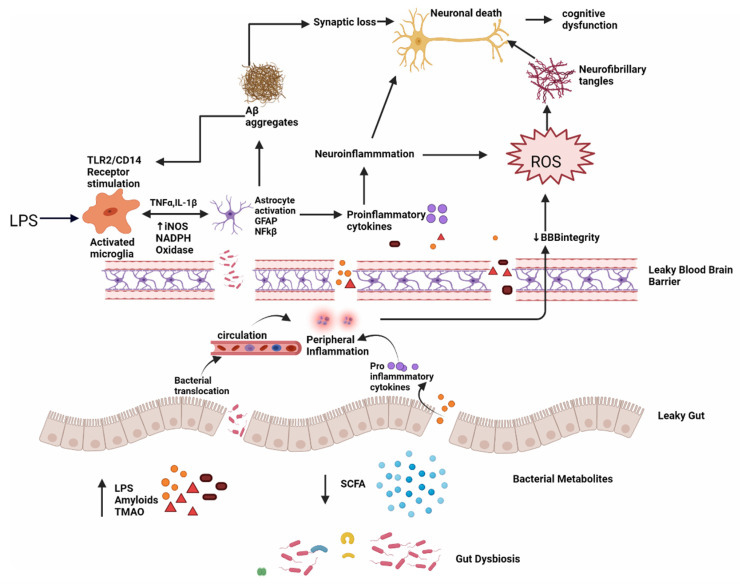
Dysbiosis and Alzheimer’s disease; intestinal permeability is harmed when gut equilibrium is disrupted by pro-inflammatory microorganisms that produce bacterial amyloids, LPS, TMAO and decrease beneficial bacterial metabolites, such as SCFA. Impairment in the gut and blood–brain barrier leads to the increased invasion of microbes into peripheral and CNS and increase production of pro-inflammatory cytokines and thus causing peripheral and central inflammation. This neuroinflammation leads to neuronal death directly and through ROS which leads to the formation of neurofibrillary tangles. LPS also acts on TLR2/TLR4 CD14 receptor on activated microglia, increasing TNF-α, IL-1β, iNOS, NADPH oxidase, and thus astrocyte activation and NF-kB activity which further promote Aβ aggregation. Aβ also acts as an agonist to the TLR4 receptor and thus promotes the vicious cycle of amyloid aggregation and ultimately neuronal death in AD.

**Figure 6 molecules-27-03402-f006:**
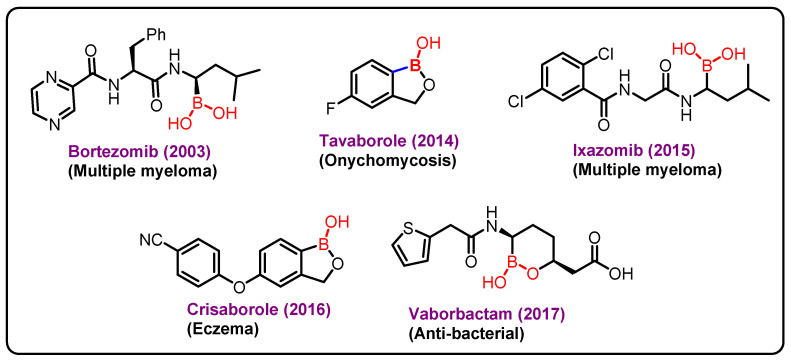
Boron-based marketed drugs.

**Figure 7 molecules-27-03402-f007:**
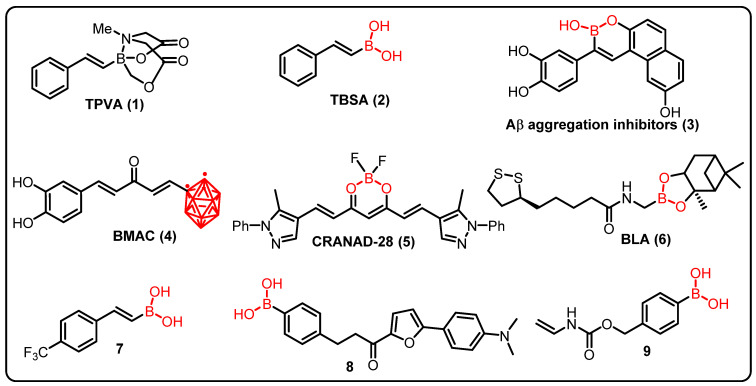
Boron containing an active compound in neuro-diseases.

**Table 1 molecules-27-03402-t001:** Various studies show alteration in gut microbiota in various neurodegenerative disorders.

Neurodegenerative Disease	Study	Experimental Subject	Control	Method	Dysbiosis/Result	Ref
**Alzhiemer’s Disease**	Liang et al. (2016)	APP/PS1 transgenic mice	C57/Bl6 wild-type (WT)	16S rRNA sequencing	↓Odoribacter, ↑Helicobacter	[94]
	Vogt et al. (2017)	Fecal samples from AD (n = 25)	sex-matched Control participants (n = 25)	16S rRNA sequencing	Firmicutes, Bifidobacterium↓, Bacteroidetes↑	[115]
	Zhang et al. (2017)	APP/PS1 transgenic male mice	Age and weight-matched littermate mice wild-type (WT)	16S rRNA sequencing	microbiota composition and diversity were perturbed and the level of SCFAs ↓in AD mice	[116]
	Cattaneo et al. (2017)	Cognitively impaired patients with (n = 40, Amy+) and with no brain amyloidosis (n = 33, Amy−)	Without brain amyloidosis and cognitive impairment (n = 10)	Microbial DNA qPCR assay	Amy+—↑pro-inflammatory cytokines (IL-6, CXCL2, NLRP3, and IL-1β) ↓anti-inflammatory cytokine (IL-10) Amy+—↓*E. rectale* and ↑ *Escherichia/Shigella*	[103]
	Zhuang et al. (2018)	Fecal samples- AD patients	age- and gender-matched cognitively normal controls	16S rRNA sequencing	At family level- ↑ Ruminococcaceae and ↓ Lachnospiraceae	[117]
	Bauerl et al. (2018)	APP/PS1 transgenic mice	C57/B16 (WT)	16S rRNA sequencing	↑ Proteobacteria and Erysipelotrichaceae	[118]
	Honarpisheh et al. (2020)	Symptomatic Tg2576 mice	age-matched littermate WT	16S rRNA sequencing	↑↑Firmicutes and Bacteroidetes ↑ Lactobacillus	[119]
**Parkinson Disease**	Cilia et al. (2020)	Fecal samples of PD pt. (n = 39)		16S rRNA sequencing	↓Roseburia (Firmicutes phylum) -worse evolution of motor, non-motor and cognitive functions ↓Ruminococcaceae and Actinobacteria- rapid worsening of global cognitive functions	[120]
	Tan et al. (2020)	Fecal samples of PD pt. (n = 104)	Control (n = 96)	16S rRNA gene sequencing	PD- ↓ SCFA (a/w poorer cognition and low BMI) and ↓ butyrate (a/w worse postural instability–gait disorder scores)	[121]
	Nishiwaki et al. (2020)	Patients with PD (n = 223)	Control (n = 137)	16S rRNA gene sequencing	PD- ↑ Akkermansia and Catabacter (genera) and Akkermansiaceae (family). ↓ Roseburia, Faecalibacterium, and Lachnospiraceae ND3007 (genera)	[122]
	Heinzel et al. 2020)	Stool sample PD pt. (n = 666)	Healthy Control		PD- ↓ Firmicutes and Faecalibacterium, ↑ Prevotella	[123]
	Shen et al. (2021)	Fifteen case–control studies		meta-analysis	PD- ↓ Prevotellaceae, Faecalibacterium, and Lachnospiraceae ↑ Bifidobacteriaceae, Ruminococcaceae, Verrucomicrobiaceae, and Christensenellaceae	[124]
	Vascellari et al. (2021)	PD patients (n = 56) (TD = Tremor Dominant-19; AR = Akinetic Rigid-23; D = Dyskinetic-14)		16S next-generation sequencing and gas chromatography-mass spectrometry	↓ Lachnospiraceae, Blautia, Coprococcus, Lachnospira, and ↑ in Enterobacteriaceae, Escherichia and Serratia linked to non-TD subtypes	[125]
**Multiple Sclerosis**	Saresella et al. (2020)	MS pt. (n = 38)	Healthy Controls (HC)		↓BA producers, ↑mucin-degrading, pro-inflammatory components BA/CA ratio was significantly ↓in MS (ratio: 0.9) compared to HC (ratio: 5; *p* < 0.0001). BA = Butyric acid CA = Caproic acid	[126]
**ALS**	Mazzini et al. (2018)	ALS patients (n = 50)	Healthy controls (n = 50)	PCR	↑E. coli and enterobacteria ↓total yeast in patients	[127]
	Gioia et al. (2020)	ALS (n = 50)	50 HC (n = 50)	PCR 16S next-generation sequencing	An unbalance between potentially protective microbial groups, such as Bacteroidetes, and other with potential neurotoxic or pro-inflammatory activity, such as Cyanobacteria, has been shown	[128]

**Table 2 molecules-27-03402-t002:** Effect of pre-biotics and probiotics on various neurological disorders.

Disease	Study	Study Design	Experimental Subject	Time	Probiotic/Prebiotic/ Psychobiotic	Effect	Ref.
**ASD**	Grimaldi et al. (2018)	RCT	prebiotic n = 13; placebo n = 13	6 wk	Prebiotic: Bimuno Galacto-oligosaccharide	↑Lachnospiraceae family, improvements in anti-social behavior, significant changes in faecal and urine metabolites	[152]
	Liu et al. (2019)	RCT	n = 80, boys with ASD, aged 7–15	4 wk	Probiotic:Lactobacillus plantarum PS128	Improve opposition/defiance behaviors	[153]
	Sanctuary et al. (2019)	RCT	n = 8, ages 2–11 with ASD and GI co-morbidities	12 wk	Bovine colostrum product (BCP)+ *Bifidobacterium infantis* (5 wk-prebiotic-probiotic combination 2 wk-washout period 5 wk-only prebiotic	↓GI symptoms and aberrant behaviors ↓ IL-13 and TNF-α production	[154]
	Wang et al. (2020)	16S rRNA gene sequencing	n = 26, ASD pt. Probiotic + FOS (n = 16) Placebo (n = 10) FOS = Fructo-oligosacchrides	30–108 days	probiotics + FOS: *B*.*infantis* Bi-26, *L*.*rhamnosus* HN001, *B*.*lactis* BL-04, *L*. *paracasei* LPC-37 and FOS	↑*Bifidobacteriales* and *B. longum* ↓ *Clostridium* ↓severity of autism and GI symptoms ↓acetic acid, propionic acid and butyric acid ↑serotonin and ↓homovanillic acid	[155]
**AD**	Akbari et al. (2016)	RCT	n = 60; probiotic (n = 30); placebo (n = 30)	12 wk	Probiotic: Lactobacillus acidophilus, Lactobacillus fermentum, Lactobacillus casei, and Bifidobacterium bifidum	Significant improvement in the MMSE score	[156]
	Kobayashi et al. (2019)	RCT	n = 121	12 wk	Bifidobacterium breve A1	Beneficial effect on the cognitive function of older people	[157]
	Ton et al. (2019)		n = 13 AD pt.	90 days	kefir synbiotic	Improves cognitive deficits, ↓markers of inflammation and oxidative stress (⋅O_2_ –, H_2_O_2_, and ONOO−, ~30%) ↑in NO bioavailability (100%)	[158]
	Kaur et al. (2020)		*App^NL-G-F^* mice	2 months	probiotic	Improved memory, ↓ plaque load and gliosis	[159]
	Bonfili et al. (2020)		3xTg-AD Eight-week-old AD male mice (n = 48)		SLAB51 probiotic: *Streptococcus thermophilus*, *Bifidobacterium lactis*, *B. lactis*, *Lactobacillus acidophilus*, *Lactobacillus helveticus*, *Lactobacillus paracasei*, *Lactobacillus plantarum*, and *Lactobacillus brevis*	Memory improvement, ↓accumulation of advanced glycation end products and ↓phosphorylated tau aggregates,	[160]
	Lee et al. (2021)		Mouse model		Prebiotic:lactulose and trehalose	Attenuated the short-term memory and the cognitive impairment of AD mice	[161]
	Cao et al. (2021)	16S rRNA gene sequencing	4-month old APP/PS1 mice	45 days	Bifidobacterium Lactis Probio-M8	↓ Aβ plaque Improve cognitive impairment	[162]
**PD**	Barichella et al. (2016)		(n = 120) PD pt.		Fermented milk containing probiotics and prebiotics	Improve constipation in PD pt.	[163]

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
