# Peer review of "The Role of Microbiome in Brain Development and Neurodegenerative Diseases"

_molecules, 2022, doi:10.3390/molecules27113402_

Round 1

Reviewer 1 Report

I congratulate the Authors, the article is very interesting. However, I would like suggest some corrections:

  1. I don't understand why Authors describe boron? The topic of article is role of microbiome in brain development and neurodegenerative diseases. However, the Authors included chapter 8. Boron as Neuroprotective agent. I think this chapter is unnecessary. Moreover, why did you describe only boron? We know many other neuroprotective agents like curcumin, glutamate blockers, statins, or COX-2 inhibitors. Some of these, e.g. curcumin have antimicrobial activity and affect microbiota. But these are not described.
  2. In lines 238-239 please add some newer references like https://pubmed.ncbi.nlm.nih.gov/35090978/ and https://pubmed.ncbi.nlm.nih.gov/34956157/ in which is presented the carcinogenic activity of bacteria.

Author Response

Reviewer 1

I congratulate the Authors, the article is very interesting. However, I would like suggest some corrections

Response:  Thank you very much for your appreciation about our review article.

Query 1: I don't understand why Authors describe boron? The topic of article is role of microbiome in brain development and neurodegenerative diseases. However, the Authors included chapter 8. Boron as Neuroprotective agent. I think this chapter is unnecessary. Moreover, why did you describe only boron? We know many other neuroprotective agents like curcumin, glutamate blockers, statins, or COX-2 inhibitors. Some of these, e.g. curcumin have antimicrobial activity and affect microbiota. But these are not described

Response:  We have discussed boron as a neuroprotective agent in this manuscript because there are very few evidence available in the literature that explains the relationship of boron with gut microbiota and its implication in certain neurological disorder.  We discussed the role of boron modulating microbiome physiology and modulating as neuroprotective agent. Boron as micronutrient- modulating microbiome and protecting neuron is the main theme of this review. As this is a special issue focusing on Boron based molecules and application, so this is the first review in the world, where we connected Boron based diet modulating microbiome colony and modulating neuro protection. As suggested by the reviewers, we have included some additional neuroprotective agents in the revised manuscript.

Query 2:  In lines 238-239 please add some newer references like https://pubmed.ncbi.nlm.nih.gov/35090978/ and https://pubmed.ncbi.nlm.nih.gov/34956157/ in which is presented the carcinogenic activity of bacteria.

Response:  As per suggestion, we have added new references in the revised manuscript [191., 192].

Reviewer 2 Report

The authors summarized the factors that influence the makeup of the microbiome, the role of gut microbiota and its metabolites in the preservation of brain functioning and the development of the aforementioned neurological illnesses. They also discussed current breakthroughs in the use of probiotics, prebiotics, and synbiotics to address neurological illnesses, the role of boron-based diet in memory, boron and microbiome relation, boron as anti-inflammatory agents and boron in neurodegenerative diseases. They are very specific and detailed. It’s very important for the research of microbiome, brain development and neurodegenerative diseases.

In conclusion, the applications prospect and research direction in the future had better be added. There are some minor grammar errors in the manuscript, the manuscript had better be carefully checked.

  1. In Line 49, the abbreviation of CNS should be defined.
  2. In Line 78, “16SrRNA” should be changed to “16S rRNA”.
  3. In Line 314, 2018 is a past time, so “investigate” should be changed to “investigated”.
  4. Figure 3 shows the structure of the tryptophan catabolites, indole and its derivatives. Therefore, the legend of figure 3 should be changed to “The structure of the tryptophan catabolites, indole and its derivatives”.
  5. Many minor errors in grammar should be carefully checked and corrected in the manuscript.
  6. 6. In line 419-420, the typeface should be identical.

Author Response

Reviewer 2

The authors summarized the factors that influence the makeup of the microbiome, the role of gut microbiota and its metabolites in the preservation of brain functioning and the development of the aforementioned neurological illnesses. They also discussed current breakthroughs in the use of probiotics, prebiotics, and synbiotics to address neurological illnesses, the role of boron-based diet in memory, boron and microbiome relation, boron as anti-inflammatory agents and boron in neurodegenerative diseases. They are very specific and detailed. It’s very important for the research of microbiome, brain development and neurodegenerative diseases.

In conclusion, the applications prospect and research direction in the future had better be added. There are some minor grammar errors in the manuscript, the manuscript had better be carefully checked

Response:  Thank you very much for your positive comments and appreciation of the work. As per suggestion, we have corrected grammatically errors in the revised manuscript. In addition, we have also added future perspective of microbiome in the conclusion part.

Query 1: In Line 49, the abbreviation of CNS should be defined.

Response:  Done

Query 2: In Line 78, “16SrRNA” should be changed to “16S rRNA”.

Response:  As suggested, 16S rRNA has changed in manuscript.

Query 3: In Line 314, 2018 is a past time, so “investigate” should be changed to “investigated”.

Response:  Corrected

Query 4:  Figure 3 shows the structure of the tryptophan catabolites, indole and its derivatives. Therefore, the legend of figure 3 should be changed to “The structure of the tryptophan catabolites, indole and its derivatives”.

Response:  According to reviewer suggestion, the legend of figure 3 has been changed in the revised manuscript.

Query 5: Many minor errors in grammar should be carefully checked and corrected in the manuscript.

Response: The grammatically errors have checked carefully in whole manuscript and corrected.

Query 6:  In line 419-420, the typeface should be identical.

Response:  Done

Round 2

Reviewer 2 Report

It can be accepted in the current version.